

# Minimizing the total waste in the one-dimensional cutting stock problem with the African buffalo optimization algorithm

Leonardo Javier Montiel-Arrieta, Irving Barragan-Vite, Juan Carlos Seck-Tuoh-Mora, Norberto Hernandez-Romero, Manuel González-Hernández and Joselito Medina-Marin

AAIyA-ICBI-UAEH, Mineral de la Reforma, Hidalgo, Mexico

## ABSTRACT

The one-dimensional cutting-stock problem (1D-CSP) consists of obtaining a set of items of different lengths from stocks of one or different lengths, where the minimization of waste is one of the main objectives to be achieved. This problem arises in several industries like wood, glass, and paper, among others similar. Different approaches have been designed to deal with this problem ranging from exact algorithms to hybrid methods of heuristics or metaheuristics. The African Buffalo Optimization (ABO) algorithm is used in this work to address the 1D-CSP. This algorithm has been recently introduced to solve combinatorial problems such as travel salesman and bin packing problems. A procedure was designed to improve the search by taking advantage of the location of the buffaloes just before it is needed to restart the herd, with the aim of not to losing the advance reached in the search. Different instances from the literature were used to test the algorithm. The results show that the developed method is competitive in waste minimization against other heuristics, metaheuristics, and hybrid approaches.

# INTRODUCTION

The one-dimensional cutting stock problem (1D-CSP) was introduced by Kantorovich in *Kantorovich (1960)*. It is one of the cutting and packing problems and is considered an NP-hard problem (*Scheithauer, 2018*). The 1D-CSP arises in many industrial applications such as shipbuilding (*Dikili & Barlas, 2011*), construction (*Benjaoran, Sooksil & Mathagul, 2017*; *Wang & Yi, 2022*), wood (*Ogunranti & Oluleye, 2016*; *Kaltenbrunner, Huka & Gronalt, 2022*), rubber mold industry *Zanarini (2017)*, and the metal industries (*Morillo-Torres et al., 2021*; *Machado et al., 2020*), to name a few. According to *Dyckhoff (1990)* the classical version of 1D-CSP can be classified into a problem of large objects named stocks and small objects or items that must be cut from the stocks. In *Wäscher, Hausner & Schumann (2007)*, the 1D-CSP falls into the single stock-size cutting stock problem

Corresponding author
Leonardo Javier Montiel-Arrieta, mo450519@uaeh.edu.mx

class where the stocks are of a fixed length while the items are of different lengths. In this article, we consider this classification for the purpose of our study. Since the 1D-CSP was introduced as a problem to deal with, several approaches have been developed to obtain optimal combinations of cutting patterns, focusing on minimizing waste. The approaches include exact algorithms, heuristics, metaheuristics and hybridizations.

The African Buffalo Optimization metaheuristic (ABO) is a novel algorithm, introduced in *Odili, Kahar & Anwar (2015)*. The ABO has been used to solve combinatorial problems like the travelling salesman problem (TSP) (*Odili et al., 2015*; *Odili & Mohmad Kahar, 2016*) and the bin packing problem (1BPP) (*Gherboudj, 2019*), where the ABO was able to obtain optimal or good solutions. An advantage of the ABO is that it requires few parameters and is easy to be implemented according to *Odili & Mohmad Kahar (2016)*. Furthermore, in *Odili et al. (2017)* is compared against other swarm intelligence algortihms in solving symetric and asymetric instances of TSP. The ABO proved to be more effective and efficient in that it was able to obtain more solutions near the optimum.

The novelty of ABO as a promising algorithm to solve combinatorial problems as well as its effectiveness and efficiency due to its simplicity motivates the study conducted in this article to continue exploring its capabilities in another sort of combinatorial problem, namely the 1D-CSP. In addition, most of the methods used to solve this problem need elaborated representations of the solutions, different stages, or the tunning of many parameters. The characteristics of the 1D-CSP which are similar to other combinatorial problems already dealt with the ABO, also guided us to adopt this algorithm since the solutions can be represented in a simple manner. However, in some preliminary experiments, we found a disadvantage pointed out by other authors which is related to continuously restarting the herd after a few iterations, leading to stagnation or to non-competitive solutions. We addressed this problem by using crossing and retention strategies at different stages of the algorithm, without adding more parameters to the ABO. The results reported in this article show that the proposed improvements to the standard version of the ABO are remarkable and competitive. Hence the main contributions of this article are on the one hand, to show the use of the ABO for solving the 1D-CSP and, on the other hand, to introduce strategies to improve its performance in solving this problem without the use of more parameters than the standard version. Additionally, we use the Ranking Order Value (ROV) method to yield discrete solutions since the ABO is aimed at handling continuous problems.

The rest of this article is organized as follows. "State of the art of 1D-CSP" contains the literature review. "Description of 1D-CSP" describes the basic concepts of 1D-CSP. "The African Buffalo Optimization algorithm" details the ABO algorithm. "Description of the ABO-1DCSP" presents the algorithm proposed. "Experiments and results" shows the results obtained. Finally, "Conclusions" presents the conclusions.

## STATE OF THE ART OF 1D-CSP

The first formal method to solve the 1D-CSP was proposed by *Kantorovich (1960)* where a model based on linear programming (LP) was introduced and the method of resolving

multipliers was used to reach to the solution. Gilmore and Gomory presented a column generation technique based on a LP formulation in *Gilmore & Gomory (1961)*, *Gilmore & Gomory (1963)* with the purpose to create feasible cutting patterns and overcoming the difficulties when applying classical procedures. Since then many other methods have been proposed based on LP to deal with the problem as it is presented in *Delorme, Iori & Martello (2016)*. In *Sá Santos & Nepomuceno (2022)* the efficiency of three methods were compared: integer linear programming, the technique of column generation, and an application of the generate and solve framework to obtain solutions. They used benchmark instances divided into five classes. Their results show that the exact approach was the only method to handle well the instances with small and medium sizes, but not for large-sized instances. Meanwhile, the method generate and solve was the only one to obtain relatively optimal solutions in almost all instances. However, the method generate and solve does not have any guarantee of convergence.

As it was pointed out above, a drawback of LP methods is that they are suitable for small or medium-sized instances of 1D-CSP, but for large-sized instances the computational cost to find the optimum solution is high. However, the performance of LP methods can be improved by heuristics like column generation. A sequential search heuristic was jointly used with LP in *Haessler (1975)*, *Haessler (1992)* to generate cutting patterns and reduce waste. Sequential heuristic procedures were later used in *Gradišar et al. (1999)*; *Cui et al. (2008)*. Lexicographic search was used in *Foerster & Wascher (2000)* to reduce the cutting patterns and for the same purpose local search was applied in *Umetani, Yagiura & Ibaraki (2003)*, *Umetani, Yagiura & Ibaraki (2006)* and *Yanasse & Limeira (2006)*.

In *Alfares & Alsawafy (2019)*, the authors implemented a heuristic model which consisted of two stages to minimize the waste and the number of stocks. They use a set of ten instances obtained from the literature and a second set from a manufacturer of office furniture. Another heuristic approach based on residual recombination was introduced in *Campello et al. (2021)*, where they merge some cutting patterns with waste to minimize the total number of stocks necessary to satisfy the demand for the items. They used different instances obtained from various works. Their results obtained were of quality within a reasonable CPU time. A method based on a greedy heuristic was developed in *Cerqueira, Aguiar & Marques (2021)* to minimize the stock. They utilize a random generator for the instances contemplating 18 classes of problems, each one with 100 instances. Their method generated solutions with a lower average of stock used. In *Lee et al. (2020)*, the authors proposed an algorithm trying to minimize the waste related to rebar in the construction industry. The algorithm was implemented in two methods. The first method called minimization by special length considers a stock with irregular length, and the second method called minimization by stock length considers a stock with standard length. Their method was applied to a case project about a commercial building. The results confirm that combining stocks with special lengths reduces the waste rate than combination by stock length. In *Vishwakarma & Powar (2021)* a mathematical model is presented, where a sustainable trim is defined as a trim loss that has a less negative economic impact on the company. The sustainable trim is used as an upper bound for all cutting patterns. If a cutting pattern has a trim loss greater than the sustainable trim, the cutting pattern is

discarded. Additionally, manpower and space are considered constraints for the problem. To validate the approach, they use real data extracted from a company transmission tower manufacturing. The model developed reduces the trim loss on average by 1.5% than other methods. Goal programming was used in *Sarper & Jaksic (2018)* to include multiple goals with soft restrictions and minimize the shortage and overage with a random demand mix of items.

Despite the fact that many heuristic approaches have been developed to tackle the 1D-CSP and its variants, one of their drawbacks is that they are designed for specific issues of the problem and cannot be implemented in a general way. For this reason metaheuristic approaches have gained popularity for solving combinatorial problems as the 1D-CSP. One of the most used evolutionary algorithms to solve the 1D-CSP is the genetic algorithm (GA) as in *Hinterding & Khan (1995)*, where two types of mappings to code the solutions into chromosomes are used to improve the performance of GA. The authors conducted their experiments on a set of five instances ranging from 20 to 126 items and they found that the use of different mappings has an impact on both the quality of results and the time to reach them. In *Liang et al. (2002)*, the authors designed a method based on Evolutionary Programming (EP) focused on minimizing both the waste and the number of stocks necessary to satisfy the orders of items. In this method, a chromosome consisted of a list of items, and it was proposed a mutation process to exchange the items with the purpose to generate new combinations of items. The proposed EP method resulted to have equal or better performance than a GA to which it was compared. The authors based their experiments on the benchmark of *Hinterding & Khan (1995)* and on a set of another five instances with a number 200, 400 and 600 of items.

In *Peng & Chu (2010b)* where, in order to minimize the trim-loss cost, two chromosomes were proposed in their method, one related to the cutting pattern and the other one to the frequency of the cutting patterns. In comparison with an EP method, the proposed hybrid method was demonstrated to be slightly better. Similarly, *Parmar, Prajapati & Dabhi (2015)* used a pair of genes where the first one is related to the frequency of the cutting pattern, while the second is for the cutting pattern itself. The modified version of GA was compared to LP, EP methods, and a two-swap algorithm on a set of 20 instances. The experiments showed that the proposed method was better in the case of multiple-stock than the LP method and better than the EP and two-swap algorithm in the case of single-stock.

Another metaheuristic used to solve the 1D-CSP is the ant colony optimization (ACO) as shown in *Levine & Ducatelle (2004)* where the first fit-decreasing method was used as a local search for the ACO. According to the authors, the hybridization and the pure ACO were comparable to GA and EP. Likewise, in *Peng & Chu (2010a)*, a tree search algorithm was applied to improve the performance of ACO. Also, in *Evtimov & Fidanova (2018)* a variant of ACO is used to solve the linear cutting stock problem. In their method, at each iteration, every ant selects randomly a stock and order. Then apply a transition probability rule searching to minimize the number of stocks. They employ a real case from a steel structure to verify the efficiency of their approach. Also, their algorithm was compared against a greedy algorithm and commercial software, obtaining positive results related to the stocks.

In *Tang et al. (2021)*, ACO is used together with the immune genetic algorithm to maximize the use of boards in the production process for wooden furniture. Also, they had to improve the pheromone update method in order to avoid the premature convergence. The method proposed was compared against others metaheuristic algorithms like GA, grey wolf optimizer, and polar bear optimization using real data provided by a furniture company. The results showed that their method can obtain a higher board utilization than the other metaheuristics. Evolutionary computation was implemented in *Chiong et al. (2008)* where each parent chromosome was generated randomly, gathering the items into groups called genes according to the length of the stock. The objective function was to minimize the number of stocks and wastage. In *Jahromi et al. (2012)*, a comparison was made between simulated annealing and tabu search to solve the 1D-CSP. They found that solutions obtained with simulated annealing have lesser waste than those related to the tabu search. However, the CPU time required to obtain optimal solutions is lesser with the tabu search.

Particle swarm optimization (PSO) has been also used to solve the 1D-CSP. In *Asvany, Amudhavel & Sujatha (2017)* a discrete PSO is proposed and compared to GA, PSO, and Cuckoo Search algorithms finding acceptable results in waste minimization, and a better performance in convergence and total material utilization. A heuristic strategy based on the use of genetic operators for PSO is implemented in *Shen et al. (2007)* to solve the 1D-CSP, and the effectiveness is demonstrated by simulation. Likewise, *Li, Zheng & Dai (2007)* made use of genetic operators as well as a hybridization of simulated annealing and general PSO to address the multistock variant of 1D-CSP. Feasible solutions were found for both limited and unlimited number of stocks. A method based on PSO was developed in *Ben Lagha, Dahmani & Krichen (2014)* for a cable manufacturer attaining comparable results against first fit-decreasing, MTP procedure and Perturbation-SAWMBS heuristic. An improvement for the artificial fish swarm algorithm was used to solve the 1D-CSP in *Cheng & Bao (2018)* yielding a better utilization rate of stock than the basic artificial fish swarm algorithm.

In *Montiel-Arrieta et al. (2022)*, the classical version of ABO was implemented to solve the 1D-CSP focusing on minimizing the number of stocks required to satisfy the number of items. They found that their method based on ABO obtains solutions very close to the best in instances with a number of items less than or equal to 60. In *Fang et al. (2023)* was presented a method based on deep reinforcement learning. Also, it was implemented a Markov decision making process to realize the cutting sequence selection of items. In addition, the parameters of the network were trained by employing the reinforcement algorithm. Their model was tested using a real steel cutting stock and a set of large scale instances generated randomly. Their approach solve effciently the 82 instances of the 3 sets. However, in some instances other approeches have better average of stock used. Meanwhile, three different methods were developed in *Srivastava et al. (2023)* based on the simplex method, PSO, and GA to reduce waste in a real-world paper industry. They consider a linear single objective with operational and technological restrictions. Their results were analyzed and compared against reported results demonstrating the efficiency of their algorithms.

The literature indicates the significant use of different approaches based on exact, heuristic, and metaheuristic methods. Even proposing hybridizations between metaheuristic approaches. In addition, the works focused on minimizing waste or the number of stocks. However, most of these methods need the use of at least two structures, additional procedures, and therefore more parameters to manage the representation of the solutions as well as the search for optimal ones. In addition, it is well-known that exact algorithms can outperform heuristics and metaheuristics methods yet they can only handle small-sized instances due to the high computational cost when addressing large-sized instances, as stated above.

In this article, we propose an algorithm for the 1D-CSP based on the ABO named ABO-1DCSP, to minimize the total waste of the cutting patterns required to satisfy the demand for items. It only requires one structure and a method to convert the continuous solutions emitted by ABO to discrete. Moreover, the standard version of the ABO avoids getting stuck on local optima by restarting the entire population of buffaloes. As it is pointed out in *Singh et al. (2020)*, however, this is done so frequently that the best solutions reached throughout the traverse of the buffaloes are lost. In order to overcome this limitation and based on crossover strategies used in PSO, like in *Yang & Li (2023)*, *Chen et al. (2018)*, *Nguyen et al. (2017)* and *Wang et al. (2008)* to avoid premature convergence, we propose to generate a new best buffalo from the best cutting patterns of the current buffaloes before restarting the herd. The purpose is that this new buffalo will guide the new reinitialized herd. Furthermore, we consider saving each best buffalo at every reinitialization of the herds because of when the termination criterion has been reached, the last best buffalo saved may not be the best of all the ones. In this way, the solution returned by the ABO-1DCSP will be the best of the saved buffaloes. These procedures are an improvement to the standard version of ABO, yielding equal or better results than other methods it is compared in this article.

## DESCRIPTION OF 1D-CSP

According to *Wäscher, Hausner & Schumann (2007)* the classical version of the 1D-CSP considers a source of long objects or stocks with different or the same length, but fixed width and rectangular shape. From the stocks are obtained small objects or items of different lengths by making straight cuts on the stocks along the width, obtaining items of rectangular shape too. The items can be arranged along the stock length in such a way they can be cut afterwards. This arrangement forms a cutting pattern and different patterns can be obtained from the stocks.

One of the main objectives of the 1D-CSP is to minimize the total waste generated by cutting all the patterns such that the demand for items is satisfied. A mathematical model given in *Jahromi et al. (2012)* for the 1D-CSP is as follows:

$$min \quad T = \sum_{j=1}^{m} tl_j \tag{1}$$

s.t.

$$\sum_{j=1}^{m} x_{ij} = n_i \forall i \qquad (2)$$

$$\sum_{i=1}^{n} x_{ij} \cdot s_i + tl_j = d_j \cdot y_j \forall j$$

$$y_j \in (0,1) \qquad (3)$$

$$x_{ij} \in \text{integer}$$

where:

- $i =$ is the $i$th item $(i = 1, ..., n)$
- $j =$ is the $j$th stock $(j = 1, ..., m)$
- $d_j =$ stock length
- $tl_j =$ stock wastes
- $s_i =$ is the length of $i$th item
- $n_i =$ is the total of items with $s_i$ length
- $T =$ sum of the cutting wastes of all cutting patterns applied
- $x_{ij} =$ integer variable, number of items with $s_i$ length that are cut from stock $j$
- $y_j =$ zero–one variable that equals to one if the stock $j$ is applied in the cutting plan otherwise, equals to zero

The objective function in Eq. (1) accounts for the total waste obtained from all the stocks $m$, which are necessary to fulfill the total demand for items according to the constraint in Eq. (2). The constraint in Eq. (3) obtain the waste of each stock selected of the cutting process.

## THE AFRICAN BUFFALO OPTIMIZATION ALGORITHM

The ABO is a swarm intelligence optimization algorithm, which was designed based on the behavior of the African buffaloes (*Odili, Kahar & Anwar, 2015*). It is focused on two sounds that African buffaloes make. The first sound, called "maa" is related to the exploitation of the current location because it is safe and has abundant pasture. The second sound, called "waa" is used to explore new places because in the current location there are dangers or the pasture is not enough (*Odili & Mohmad Kahar, 2016*). The main steps of the ABO are as follows:

1. Set the objective function.
2. Randomly generate a population of $N$ buffaloes.
3. Update the fitness of each buffalo according to Eq. (4).

   $$m_{k+1} = m_k + lp1(bgmax - w_k) + lp2(bpmax_k - w_k) \qquad (4)$$

   In Eq. (4), $m_k$ represents the exploitation and $w_k$ stands for the exploration of the $k$th buffalo k =1 $(2, ..., N)$, while $lp1$ and $lp2$ are learning factors. The $bgmax$ is the best buffalo of the herd. Meanwhile, the $bpmax_k$ is the best location of each buffalo along its traverse.

4.  Update the location of each buffalo using Eq. (5).

$$w_{k+1} = \frac{w_k + m_k}{\pm\lambda} \tag{5}$$

According to *Odili, Mohmad Kahar & Noraziah (2017)* $\lambda$ could take values from 0.1 to 2. If the value of lambda is low, it will promote exploration; otherwise, it will encourage exploitation.

5.  If there is a change in the bgmax value after updating the fitness of all the buffaloes, then go to step 6. Otherwise, go to step 2 to reinitialize the herd.

6.  If the stop criterion is reached, go to step 7. Otherwise, go to step 3.

7.  Return bgmax as the best solution.

It can be noticed that the ABO tends to avoid stagnating because the best buffalo of the herd is continuously updated. In addition, the movement of exploration takes into account the best location of each buffalo as well as the best buffalo of the whole herd, giving the algorithm a memory property.

As the standard version of ABO reinitializes the entire herd if the bgmax is not updating, this misses the advance gained in the search for the best solution. Hence in the ABO-1DCSP, we propose the following:

1.  Generate a bgmax based on the last buffaloes before reinitialization, taking the best cutting patterns from them.

2.  Since the last bgmax found will be replaced by the newly generated bgmax, it will be saved and used later to obtain the best or global solution when the termination condition is reached.

## DESCRIPTION OF THE ABO-1DCSP

This section describes the proposed algorithm to address the 1D-CSP based on the ABO algorithm. Firstly, it is explained how the solutions are represented, then the fitness function is given to evaluate the solutions and finally the steps of the algorithm are detailed, which includes a procedure to make discrete the solutions.

### Representation of the solutions

The algorithm ABO-1DCSP searches for a solution with minimal waste. A solution or buffalo consists of a linear arrangement of items from which a number of cutting patterns are generated by summing the lengths of the items from the leftmost item to the rightmost one. A cutting pattern is made of a group of consecutive items such that the sum of their lengths does not exceed the length of the stock assigned in turn. Any remainder of the stock not used is considered waste.

In order to explain how to represent a solution for the ABO-1DCSP, we use the instance of Table 1 as an example. This instance has four items with different lengths and the same demand. Also, it considers an unlimited number of stocks with a length of 65. In Fig. 1, some examples of solutions or buffaloes are shown, where each buffalo contains all the items of the instance of Table 1, including the demand for each of them.

In Fig. 2 it is shown that the formation of cutting patterns begins from the left side of the arrangement to the right. If the sum of the length of a new item exceeds the length of

**Table 1** The example instance with a stock length of 65.

| Item | Length | Demand |
|---|---|---|
| Item 1 | 40 | 2 |
| Item 2 | 30 | 2 |
| Item 3 | 25 | 2 |
| Item 4 | 15 | 2 |

**Figure 1** Solutions representation or buffaloes for the 1D-CSP.

**Figure 2** Cutting patterns formation for each buffalo of Fig. 1.

the stock, then the new item will be the first of the new stock and the sum of lengths is reinitialized. This process continues with the remaining items of the buffalo and finally, it is determined the total waste, the number of stocks used as well as the number of stocks with waste.

From Fig. 2, it can be seen that buffaloes 1 and 2 have the same waste. However, the number of stocks used and the number of stocks with waste in buffalo 1 and 2 is lesser than in buffalo 3, whereas buffalo 2 used the whole length of stock 1 and 2.

## Fitness function

In order to evaluate the solutions quality and to determine the best buffalo of the herd throughout the iterations, a fitness function based on the total waste generated by the

arrangement of the items of each buffalo is proposed. The total waste of each buffalo is computed as indicated in Eq. (6) where the partial waste on each stock $j$ is determined as the difference between the length of the stock $l_j$ and the length of the corresponding cutting pattern $lp_j$ such that the sum of all of the partial wastage yields the total waste $w$ of total stocks used $ts$.

$$w = \sum_{j=1}^{ts} l_j - lp_j \tag{6}$$

where:

- $w$ = the total waste generated by cutting patterns.
- $l_j$ = the length of stock $j$.
- $lp_j$ = the length of cutting pattern.
- $ts$ = total number of stocks used to accomplish the order of items.

## ABO-1DCSP

This section presents how the algorithm ABO-1DCSP works to solve the 1D-CSP, based on the ABO. The steps of ABO-1DCSP are detailed in Algorithm 1.

According to Algorithm 1, is needed to set the parameters $lp1$, $lp2$, $\lambda$ , the number of iterations to stop the algorithm ($k$), the number of buffaloes ($nb$), and the number of iterations needed to reinitialize the herd ($q$). In the ABO-1DCSP, the herd of buffaloes is restarted after every $q$ iterations as long as the $bgmax$ has not been updated; otherwise, the search continues in such a way the count of iterations to restart the herd is reinitialized with the aim to allow the current best buffalo to guide the herd to find better cutting patterns. Additionally, in the algorithm ABO-1DCSP is introduced a procedure of crossover to build a $bgmax$, namely $bgmax_B$, when a restart of the herd is needed and the current $bgmax$ has not been updated after $q$ iterations. The $bgmax_B$ is obtained from the best cutting patterns of the buffaloes of the last iteration and the current $bgmax$ is saved in order to be used when $k$ iterations are reached. This crossing process has not been implemented with the ABO. While other investigations have proposed dividing the herd into efficient and non-efficient buffaloes as seen in *Singh et al. (2020)*. In addition, crossing processes have already been designed with swarm intelligence algorithms such as the PSO, as observed in *Yang & Li (2023)*, *Chen et al. (2018)*, *Nguyen et al. (2017)* and *Wang et al. (2008)*.

After setting the values of the parameters, the first step of our algorithm consists of creating randomly an initial population of $nb$ buffaloes by using the function *CreateBuffaloesRandom* (line 1 of Algorithm 1). Then, the $bgmax$ is determined based on the evaluation of the fitness function for each buffalo. This is done with the *Searchbgmax* function displayed on line 2 of Algorithm 1.

Once the initialization of the algorithm has been carried out, the process continues with updating the location of each buffalo. The *UpdateBuffaloes* function uses the Eqs. (4) and (5) to update the buffaloes, yielding the $mb$ and $wb$, correspondingly. In all of the instances tested in this article the lengths of the items as well as of the stocks are discrete and therefore, we needed discrete solutions. Since the values obtained with Eq. (4) and

**Input:** instance, stock_length, number_buffaloes(nb), $\lambda$, iterations (k), lp1, lp2,
      number_iterations_restart(q)

**Output:** best_bgmax

1  buffaloes = CreateBuffalosRandom(instance, stock_length, nb);

2  bgmax = Searchbgmax(buffaloes, stock_length);

3  j = 1;

4  i = 1;

5  **while** $i \leq k$ **do**

6      buffaloes = UpdateBuffaloes(buffaloes, stock_length, nb, $\lambda$, lp1, lp2, instance, bgmax);

7      bgmaxupdated, bgmax = VerifyUpdatebgmax(buffaloes, bgmax, stock_length);

8      **if** *bgmaxupdated == False and i % q == 0* **then**

9         list_bgmax[j] = bgmax;

10         j = j + 1;

11         $\text{bgmax}_B$ = Generatebgmax(buffaloes, instance, stock_length);

12         buffaloes = CreateBuffalosRandom(instance, stock_length, nb);

13         bgmax = $\text{bgmax}_B$;

14         i = i + 1;

15      **else**

16         i = i + 1;

17      **end**

18  **end**

19  list_bgmax[j] = bgmax;

20  best_bgmax = SearchBestbgmax(list_bgmax);

<b>Algorithm 1:</b> ABO-1DCSP.

Eq. (5) are continuous, the ROV method was used as in *Liu et al. (2008)* to obtain discrete values.

In order to explain the ROV was used, let us consider the instance shown in Table 1 and Fig. 3. Firstly, it is necessary to sort the items by the length in ascending order and number them as shown in Fig. 3A. Then, suppose a new *wb* is obtained with Eq. (5) as the one presented in Fig. 3B. The next step consists in sorting the values of the new *wb* in ascending order as shown in Fig. 3C and let *j* and *i* be the new and old index of the sorted values. Therefore, the *ith* position of the discretized *wb* will be filled with the length of the *jth* item of the list of the sorted items. As an example, consider the new $j = 1$ position and the corresponding old $i = 3$ position of Fig. 3C. Thus, the $i = 3$ position of Fig. 3D is filled with the length of the $j = 1$ item of Fig. 3A. The process is repeated with the rest of the values of Fig. 3C to generate the new arrangement of items presented in the Fig. 3D.

After each *wb* has been updated and discretized, they are evaluated with the fitness function and the best buffalo of the new herd is determined. Consequently, the algorithm uses the *VerifyUpdatebgmax* function to check if *bgmax* was updated after all buffaloes were reallocated.

| List of items | | New Wb Obtained | | New Wb Sorted | | | New Wb Discretized | |
|:---:|:---:|:---:|:---:|:---:|:---:|:---:|:---:|:---:|
| **Position** | **Length** | **Position** | **Value** | **Old Position ($i$)** | **New Position ($j$)** | **Value** | **Position** | **Length** |
| 1 | 15 | 1 | 41.5 | 3 | 1 | 14.9 | 1 | 40 |
| 2 | 15 | 2 | 31.9 | 8 | 2 | 15.6 | 2 | 30 |
| 3 | 25 | 3 | 14.9 | 4 | 3 | 23.7 | 3 | 15 |
| 4 | 25 | 4 | 23.7 | 7 | 4 | 25.3 | 4 | 25 |
| 5 | 30 | 5 | 39.4 | 6 | 5 | 29.1 | 5 | 40 |
| 6 | 30 | 6 | 29.1 | 2 | 6 | 31.9 | 6 | 30 |
| 7 | 40 | 7 | 25.3 | 5 | 7 | 39.4 | 7 | 25 |
| 8 | 40 | 8 | 15.6 | 1 | 8 | 41.5 | 8 | 15 |
| **A** | | **B** | | **C** | | | **D** | |

**Figure 3** **Explanation of ROV.** (A) The items of instance are sorted in ascending way according to their length. (B) They are the new values obtained related to the wb. (C) The values of the New wb are ordered in ascending order (new position), and the old position is placed. (D) They are the discretized values of the new wb.

In our algorithm, each time $q$ iterations are reached and the $bgmax$ has not been updated, it is avoided this $bgmax$ leads a new herd under the assumption that this buffalo ($bgmax$) is not suitable to perform such a task since it failed to lead the past herds to find a better solution. However, this $bgmax$ is not discarded at all but is saved to be used when the $k$ iterations has been fulfilled to stop the search.

Following with the Algorithm 1, when the $bgmax$ has not been updated after $q$ iterations, it is saved in *list_bgmax* and the procedure of crossover is performed by *Generatebgmax* function to obtain the $bgmax_B$ from the best cutting patterns of the herd in the last iteration. This procedure is carried out since it was considered that the information about the last location of the buffaloes should not be missed despite it was not found a better solution than the current $bgmax$ that has lead the herd until the $q$-th location. This information about the location of the buffaloes is preserved in the $bgmax_B$ made up from the best cutting patterns of the last herd.

To explain how the function *Generatebgmax* generates the $bgmax_B$, let us consider the test instance of the Table 1 as an example and the buffaloes shown in Fig. 4A as the herd in the last iteration. Hence, the buffaloes are sorted in ascending order by their total waste. Then the cutting patterns of each buffalo are also sorted in ascending order by the waste of each pattern in such a way that the patterns are arranged from the left to the right in each buffalo as it can be seen in Fig. 4B. Once the buffaloes and their cutting patterns are sorted, the process continues with the selection of the best cutting patterns to form the $bgmax_B$. The first pattern to be selected is the first one (from the left to the right) of the first buffalo since this buffalo has the minimum total waste and the pattern the minimum waste. This

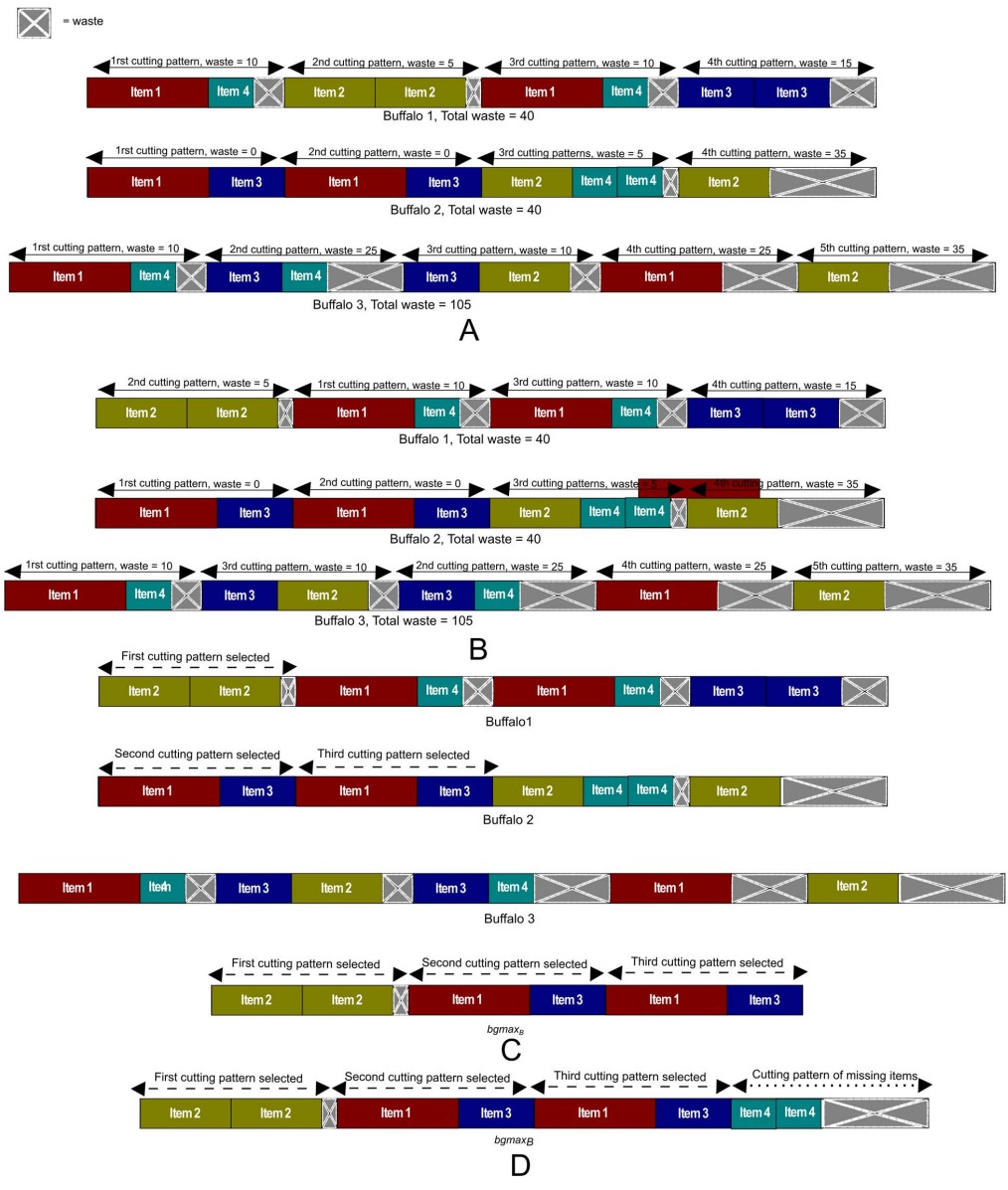

**Figure 4  The process to get the *bgmax_B*.** (A) The current herd of example from a *bgmax_B* will be generated. (B) The cutting patterns of each buffalo and buffaloes are sorted ascending according to the number of waste. (C) The best cutting patterns of the current herd are chosen for the *bgmax_B*. (D) The items missing from the instance are added ascending at the end of the *bgmax_B*.

first buffalo will be the reference for the search, and the pattern will be the first one of the *bgmax_B* as it is shown in Fig. 4C. The demand for each of the items in the selected pattern is discounted in the list of items, and each selected pattern is discarded from the buffalo that belongs. The next step is to check if the following pattern of the reference buffalo has zero waste. Otherwise, it will skip to the next buffalo. It can be seen with the first buffalo of Fig. 4C where the first pattern is selected, but the next pattern has not zero waste therefore it goes to the next buffalo. If the first pattern of the next buffalo can be

selected, such a buffalo becomes the new reference to continue the search. Otherwise, the pattern is discarded from the buffalo, and it goes to the next buffalo to continue the search until the first pattern of a buffalo can be selected.

Following this process, it may occur that in the list of items there are items that do not belong to a pattern, that is to say, they do not form a pattern. Therefore, these items are arranged in ascending order by the length and added to the end of the $bgmax_B$ to complete the buffalo as shown in Fig. 4D.

In this way, the $bgmax_B$ will guide the new herd generated randomly with the function *CreateBuffaloesRandom* and a new set of $q$ iterations is initialized. Completed all the $k$ iterations, the function *SearchBestbgmax* of Algorithm 1 searches for the best solution *best_bgmax* in *list_bgmax*.

## EXPERIMENTS AND RESULTS

The algorithm ABO-1DCSP has been programmed with Python language, version 3.7 and executed in a Intel® Core™ i7-6700HQ CPU 2.60 GHz computer with 8 GB of RAM, under Windows 10 Home Single Language.

We selected 130 instances to test the performance of the ABO-1DCSP. Each instance consists of several items of rectangular dimensions with a fixed width and variable length, and only one stock of rectangular dimensions is used on each instance. That is, there are no stocks of different lengths, but it is allowed to use the necessary number of stocks to fulfill the demand for the items. Firstly, we chose a set of ten instances named from 1a to 10a obtained from *Liang et al. (2002)*, which were used to test algorithms in some studies, and we found them suitable to evaluate the performance of our algorithm and to be compared with the results of those algorithms. Secondly, the next 80 instances were used in *Falkenauer (1996)* and were obtained from http://people.brunel.ac.uk/~mastjjb/jeb/orlib/binpackinfo.html. The last 40 instances were used in *Scholl, Klein & Jürgens (1997)* and were obtained from set 1 y 2 of https://www.euro-online.org/websites/esicup/data-sets/. To the best of our knowledge, there is no optimal values known for all these instances. However the theoretical lower optimum bound can be easily obtained. The information on the number of items and length of stock of instances is shown in Table 2.

In regard to the instances obtained from *Liang et al. (2002)*, the ABO-1DCSP is compared with eight algorithms. Seven of these algorithms, namely those from *Liang et al. (2002)*, *Levine & Ducatelle (2004)*, *Chiong et al. (2008)*, *Peng & Chu (2010a)*, *Peng & Chu (2010b)* and *Parmar, Prajapati & Dabhi (2015)*, are population-based metaheuristics while the algorithm from *Alfares & Alsawafy (2019)* is a heuristic approach. We have taken only the values reported in the articles related to the comparisons with instances 1a to 10a. The authors of these works used fully or partially the same set of instances. Nevertheless, the performance measures which are compared are not the same for all the algorithms as it can be seen in Table 3. For example, ABO-1DCSP under the average stock with waste is compared against *Chiong et al. (2008)* and *Liang et al. (2002)* since in this works the focus was on this performance metric.

Although the main objective of the ABO-1DCSP algorithm is to find the solution with the minimal waste, we consider additional performance measures like the number of

**Table 2  Description of instances.**

| | Instance | Number of items | Items length range, Minimum–Maximum | Length of stock |
|---|---|---|---|---|
| | 1a | 20 | 3–10 | 14 |
| | 2a | 50 | 3–10 | 15 |
| | 3a | 60 | 3–10 | 25 |
| | 4a | 60 | 5–12 | 25 |
| *Liang et al. (2002)* | 5a | 126 | 1,050–2,350 | 4,300 |
| | 6a | 200 | 21–47 | 86 |
| | 7a | 200 | 22–64 | 120 |
| | 8a | 400 | 22–67 | 120 |
| | 9a | 400 | 21–67 | 120 |
| | 10a | 600 | 21–67 | 120 |
| | u120_00-u120_19 | 120 | 20–120 | 150 |
| *Falkenauer (1996)* | u250_00-u250_19 | 250 | 20–120 | 150 |
| | u500_00-u500_19 | 500 | 20–120 | 150 |
| | u1000_00-u1000_19 | 1000 | 20–120 | 150 |
| *Scholl, Klein & Jürgens (1997)* | Set 1 of Scholl | 50–500 | 2–100 | 100–120 |
| | Set 2 of Scholl | 50–500 | 16–494 | 1,000 |

**Table 3  Algorithms related to performance measures.**

| Algorithm | Reference | Performance measures | | |
|---|---|---|---|---|
| | | Average waste | Average stock used | Average stock used with waste |
| LLA | *Alfares & Alsawafy (2019)* | ✓ | ✓ | |
| SMBEP | *Chiong et al. (2008)* | ✓ | ✓ | ✓ |
| HACO | *Levine & Ducatelle (2004)* | | ✓ | |
| Pure ACO | *Levine & Ducatelle (2004)* | | ✓ | |
| EP | *Liang et al. (2002)* | ✓ | ✓ | ✓ |
| MACO | *Peng & Chu (2010a)* | ✓ | ✓ | |
| HMCGA | *Peng & Chu (2010b)* | ✓ | ✓ | |
| GA | *Parmar, Prajapati & Dabhi (2015)* | ✓ | | |

stocks used and the number of stocks with waste which are easily determined once the best solution has been found. This is done under the fact that the 1D-CSP we address consists of stocks of a unique length and the demand of items must be met strictly, and therefore the miminimization of total waste implies the minimization of stocks used.

For all of the instances and after preliminary experiments, the chosen population size of buffaloes was 90, and the chosen number of iterations to search for the best solution is set to 440, where both quantities are in the range suggested in *Odili et al. (2022)*. The values of the learning factors *lp1* and *lp2* were set to 0.3 and 0.6, respectively, based on *Gherboudj (2019)*. The value of λ was set to 1 because it was desired to have a balance between exploitation and exploration as was pointed out in *Odili, Mohmad Kahar & Noraziah (2017)*. Moreover, after the preliminary experiments, it was determined that 10 iterations are needed to restart

**Table 4 Experimental parameters setting.**

| ABO-1DCSP | | SMBEP | | HACO | | Pure ACO | | EP | |
|---|---|---|---|---|---|---|---|---|---|
| Parameter | Value | Parameter | Value | Parameter | Value | Parameter | Value | Parameter | Value |
| Buffaloes | 90 | Generations | 1,000–2,000 | nants | No. of items | nants | No. of items | Population | 75 |
| lp1 | 0.3 | | | Value of fitness function | 2 | Value of fitness function | 2 | Tournament size | 10 |
| lp2 | 0.6 | | | $\beta$ | 1–2 | $\beta$ | 2–10 | Generations | 50–20,000 |
| λ | 1 | | | $\gamma$ | 1 | $\gamma$ | 500/total of items | | |
| q | 10 | | | $\rho$ | 0.1–0.9 | $\rho$ | 0.95 | | |
| k | 440 | | | Evolutions | 20,000 | Evolutions | 100,000 | | |
| | | | | bins | 4 | | | | |
| Total No. of runs | 50 | | 10 | | 50 | | 50 | | 50 |

the herd if the best solution found is not updated. The ABO-1DCSP algorithm was run 50 times on each instance as was done in *Liang et al. (2002)* and *Levine & Ducatelle (2004)*.

Table 4 shows the different parameters reported by the approaches that use the set of *Liang et al. (2002)* in their experiments compared against the ABO-1DCSP. It should mention that four of the eight methods show the parameters that they used for their experiments. It can see that almost all the methods perform 50 executions of their algorithm except for the SMBEP approach to obtain the results that they show.

Apart from comparing the performance of the ABO-1DCSP with the previous algorithms, the relative percentage deviation (RPD) was obtained as shown in *Madenoğlu (2021)* and *Ruiz, Vallada & Fernández-Martínez (2009)* with Eq. (7), where BOV is the Best Obtained Value by each algorithm, and BKV is the Best Known Value for instances 1a to 10a. After calculating the RPD it was carried out a non-parametric Friedman test using the language R version 4.2.0 for performance measure of Average Stock used and Average Waste, with the aim to determine whether there is a statistically significant difference between the results obtained among all of the algorithms based on their RPD values as in *Derrac et al. (2011)* and *Serna et al. (2021)*.

$$RPD = \frac{BOV - BKV}{BOV} \times 100 \tag{7}$$

Table 5 presents the results for the average minimal waste. The best average minimal waste obtained in instances 1a to 10a is indicated with a *. As it can be observed, for the instances 1a, 2a, 3a, 4a, 7a, the ABO-1DCSP reaches the best average minimal waste like other algorithms, whereas for the instance 9a, it reaches the best average minimal waste. In the instance 5a, it obtains a lesser or equal average waste in comparison with the methods LLA, EP, HMCGA, and MACO, but not for the SMBEP and GA. For the instance 6a, the ABO-1DCSP produces less waste compared to the other methods except for HMCGA and MACO. In the instance 8a, the ABO-1DCSP yields the less average waste against LLA, EP, and GA methods. In the instance 10a, the ABO-1DCSP generated less average waste compared to almost all approaches except for LLA and SMBEP.

**Table 5   Average waste.**

| Instance | BKV | ABO-1DCSP | LLA | SMBEP | EP | GA | HMCGA | MACO |
|---|---|---|---|---|---|---|---|---|
| 1a | 3 | 3[*] | 3[*] | 3[*] | 3[*] | 3 | 3[*] | 3[*] |
| 2a | 13 | 13[*] | 13[*] | 13[*] | 13[*] | 14.5 | 13[*] | 13[*] |
| 3a | 0 | 0[*] | 0[*] | 0[*] | 0[*] | 2.5 | 0[*] | 0[*] |
| 4a | 11 | 11[*] | 11[*] | 11[*] | 11[*] | 11[*] | 11[*] | 11[*] |
| 5a | 10,850 | 11,450 | 11,450 | 11,370 | 11,966 | 10,850[*] | 11,966 | 11,966 |
| 6a | 103 | 109.88 | 275 | 240.6 | 309.4 | 330.9 | 103[*] | 103[*] |
| 7a | 84 | 84[*] | 84[*] | 84[*] | 189.6 | 327.6 | 264 | 264 |
| 8a | 212 | 320 | 332 | 308 | 788 | 547.95 | 212[*] | 212[*] |
| 9a | 142 | 142[*] | 382 | 250 | 730 | 673.8 | 334 | 334 |
| 10a | 130 | 274 | 130[*] | 190 | 1037.2 | 662.5 | 490 | 490 |

Notes.
[*] Best obtained average minimal waste.

**Table 6   Average RPD and friedman test for waste.**

| Algorithm | ABO-1DCSP | LLA | SMBEP | EP | GA | HMCGA | MACO | *p* value |
|---|---|---|---|---|---|---|---|---|
| Average RPD: | 9.78 | 16.67 | 16.77 | 37.28 | 47.41 | 20.84 | 20.84 | 0.0047 |
| Rank | 1 | 2 | 3 | 5 | 6 | 4 | 4 | |

The results of the Average RPD and non-parametric Friedman test related with the Table 5 for the average minimal waste are presented in Table 6. As shown the ABO-1DCSP is ranked in the first position according to Average RPD, displaying the high effectiveness of the ABO-1DCSP to minimize waste against the algorithms compared. Meanwhile the *p* value related to Friedman test is 0.0047, as observed in the last column, which is lesser than the significant level of 0.05 and allows us to reject the null hypothesis that all algorithms behave statistically similarly. The *p* value indicates significant differences in the performance of all algorithms according to parameter waste in relation to the waste produced.

The comparison of ABO-1DCSP against other algorithms under the average stock used is presented in Table 7. As can be seen from Table 7, the ABO-1DCSP from instances 1a to 5a uses an equal or less quantity of average stock used even though ABO-1DCSP will consider an objective function based on the total waste. Meanwhile, the others approaches consider the stock used in their objective function. For instance 6a, the ABO-1DCSP use less stock than other methods except for the PureACO and HACO. In instance 7a, the ABO-1DCSP obtained a lower or equal average stock used against all algorithms except HMCGA and MACO. For instance 8a, the ABO-1DCSP gets a lower average stock used against LLA, EP and Pure ACO algorithms. For instance 9a, the ABO-1DCSP obtains less stock used than algorithms LLA, SMBEP, EP, and Pure ACO. In the last instance, the ABO-1DCSP obtained less stock used than EP, Pure ACO, HMCGA, and MACO.

In the same way, a non-parametric Friedman test was applied with RPD values related to Table 7 associated with the average stock used, except for values of algorithms HACO and Pure ACO because they do not have results in instances 1a to 5a. As shown in Table 8

**Table 7** Average stock used.

| Instance | BKV | ABO-1DCSP | LLA | SMBEP | EP | Pure ACO | HACO | HMCGA | MACO |
|---|---|---|---|---|---|---|---|---|---|
| 1a | 9 | 9[*] | 9[*] | 9[*] | 9[*] | – | – | 9[*] | 9[*] |
| 2a | 23 | 23[*] | 23[*] | 23[*] | 23[*] | – | – | 23[*] | 23[*] |
| 3a | 15 | 15[*] | 15[*] | 15[*] | 15[*] | – | – | 15[*] | 15[*] |
| 4a | 19 | 19[*] | 19[*] | 19[*] | 19[*] | – | – | 19[*] | 19[*] |
| 5a | 53 | 53[*] | 53[*] | 53[*] | 53.12 | – | – | 53[*] | 53[*] |
| 6a | 79 | 79.08 | 81 | 80.6 | 81.4 | 79[*] | 79[*] | 79.1 | 79.1 |
| 7a | 67.3 | 68 | 68 | 68 | 68.88 | 69 | 68 | 67.3[*] | 67.3[*] |
| 8a | 143 | 144.9 | 145 | 144.8 | 148.8 | 146 | 143[*] | 144.8 | 144.8 |
| 9a | 149 | 150 | 152 | 150.9 | 154.9 | 151 | 149[*] | 149.4 | 149.4 |
| 10a | 215 | 217.2 | 216 | 216.5 | 223.56 | 218.9 | 215[*] | 219.8 | 219.8 |

**Notes.**
 [*]Best obtained average stock.

**Table 8** Average RPD and friedman test for parameter stock used.

| Algorithm | ABO-1DCSP | LLA | SMBEP | EP | HMCGA | MACO | p value |
|---|---|---|---|---|---|---|---|
| Average RPD: | 0.41 | 0.73 | 0.62 | 1.7 | 0.38 | 0.38 | 0.0034 |
| Rank | 2 | 4 | 3 | 5 | 1 | 1 | |

the ABO-1DCSP is ranked in the second position in accordance with Average RPD, demonstrating the high competence of the ABO-1DCSP to minimize stock used against the algorithms compared, staying very close to the first place. The result of the Friedman test was 0.0034, as observed in the last column of Table 8, which is less than the significant level of 0.05 to reject the null hypothesis that all algorithms perform statistically similarly. The p value shows considerable differences between algorithms under the parameter stock used and the effectiveness of ABO-1DCSP to minimize the stock used.

Table 9 shows the comparison of ABO-1DCSP under the parameter of the stock with waste against algorithms EP and SMBEP. It is observed that ABO-1DCSP compared against SMBEP algorithm obtain the same or minus average stocks used with waste from instance 1a to 6a, but not with the other instances. Meanwhile, the ABO-1DCSP had the same or less average stocks used with waste than EP in instances 3a, 6a, 8a, 9a, and 10a, but not in instances 1a, 2a, 4a, 5a, and 7a.

We can not perform a non-parametric Friedman test with RPD values related to Table 9 associated with the stock used with waste because it has no requirements to perform it. However, the ABO-1DCSP obtained values closed to the best average in almost all instances and positioned in the second position, conforming to Average RPD as shown in Table 10.

Table 11 shows the comparison of the ABO-1DCSP against the theoretical lower optimum bound ($N_{min}$) of stock used shown in *Alfares & Alsawafy (2019)* for the *Liang et al. (2002)* instances. We take the amount of stock used from each method shown in Table 7 to obtain the percentage difference against $N_{min}$. It can be appreciated the HMCGA and MACO approaches obtained a stock lower than $N_{min}$ in instance 7a, so their percentages are negative. As can be seen, the ABO-1DCSP reaches the $N_{min}$ in 5 out of the 10 instances.

**Table 9  Average stock used with waste.**

| Instance | BKV | ABO-1DCSP | SMBEP | EP |
|---|---|---|---|---|
| 1a | 2 | 2.3 | 2.8 | 2* |
| 2a | 4 | 4.48 | 4.7 | 4* |
| 3a | 0 | 0* | 0* | 0* |
| 4a | 1.02 | 2.1 | 3.2 | 1.02* |
| 5a | 22.8 | 23.48 | 27.1 | 22.8* |
| 6a | 24.8 | 24.8* | 26.5 | 29.96 |
| 7a | 6.6 | 8.34 | 6.6* | 7.48 |
| 8a | 27.4 | 33.34 | 27.4* | 56.24 |
| 9a | 17.6 | 23.22 | 17.6* | 48.54 |
| 10a | 11.4 | 33.52 | 11.4* | 73.06 |

**Table 10  Average RPD for parameter stock used with waste.**

| Algorithm | ABO-1DCSP | SMBEP | EP |
|---|---|---|---|
| Average RPD: | 20.69 | 13.37 | 22.82 |
| Rank | 2 | 1 | 3 |

In the instance 5a, the ABO-1DCSP was no more than 4% of $N_{min}$. Meanwhile, in the 6a, 8a, and 10a instances, the ABO-1DCSP was just above 1% of $N_{min}$, and in the instance 9a, the ABO-1DCSP is 0.67% above $N_{min}$ to complete the demand for items. It is essential to mention that our algorithm focuses on a single objective in the search for better solutions. The method ABO-1DCSP employs an objective function that only seeks to minimize waste. The algorithm SMBEP consider the amount of waste related in the chromosomes. However, the algorithms PureACO and HACO use a objective function focused on taking advantage of the stock. Meanwhile, the methods HMCGA and MACO applied a objective función to reduce the cost stock. Contrarily, the approach EP implement a objective function that try to minimize waste and stock with waste. On the other hand, the heuristic LLA considers use two objective functions that involve minimizing waste and the number of stocks necessary to satisfy the demand for items.

Other tests were performed with instances of *Falkenauer (1996)* to compare the effectivenness of ABO-1DCSP under the parameter stock used against the $N_{min}$ reported in *Gherboudj (2019)*. All the instances considered only a single type of stock with a length of 150. The instances are classified into four classes according to the number of items, which are 120, 250, 500, and 1,000 items.

Table 12 shows that the ABO-1DCSP equals the $N_{min}$ in 14 out of the 20 instances. In the instances u120_03 and u120_17, the ABO-1DCSP is 0.2% above $N_{min}$. For instances u120_09 and u120_12, the ABO-1DCSP is between 1 and 1.8% above $N_{min}$ to complete the order of items. For the instances, u120_08 and u120_19, the ABO-1DCSP is 1.99% above $N_{min}$ to satisfy the demand for items.

Table 13 shows that the ABO-1DCSP reaches the $N_{min}$ in 8 out of the 20 instances. In the instances u250_00, u250_02, u250_04, u250_05, u250_08, u250_11, u250_12, u250_15

**Table 11 Comparison of $N_{min}$ stock used of instances of *Liang et al. (2002)*.**

| Instance | $N_{min}$ | % ABO-1DCSP > $N_{min}$ | % LLA > $N_{min}$ | % SMBEP > $N_{min}$ | % EP > $N_{min}$ | % Pure ACO > $N_{min}$ | % HACO > $N_{min}$ | % HMCGA > $N_{min}$ | % MACO > $N_{min}$ |
|---|---|---|---|---|---|---|---|---|---|
| 1a | 9 | 0 | 0 | 0 | 0 | – | – | 0 | 0 |
| 2a | 23 | 0 | 0 | 0 | 0 | – | – | 0 | 0 |
| 3a | 15 | 0 | 0 | 0 | 0 | – | – | 0 | 0 |
| 4a | 19 | 0 | 0 | 0 | 0 | – | – | 0 | 0 |
| 5a | 51 | 3.92 | 3.92 | 3.92 | 4.15 | – | – | 3.92 | 3.92 |
| 6a | 78 | 1.38 | 3.84 | 3.33 | 4.35 | 1.28 | 1.28 | 1.41 | 1.41 |
| 7a | 68 | 0 | 0 | 0 | 1.29 | 1.47 | 0 | −1.02 | −1.02 |
| 8a | 143 | 1.32 | 1.39 | 1.25 | 4.05 | 2.09 | 0 | 1.25 | 1.25 |
| 9a | 149 | 0.67 | 2.01 | 1.27 | 3.95 | 1.34 | 0 | 0.26 | 0.26 |
| 10a | 215 | 1.02 | 0.46 | 0.69 | 3.98 | 1.81 | 0 | 2.23 | 2.23 |

**Table 12 Comparison of $N_{min}$ stock used of instances u120 of *Falkenauer (1996)*.**

| Instance | $N_{min}$ | ABO-1DCSP | % ABO-1DCSP > $N_{min}$ |
|---|---|---|---|
| u120_00 | 48 | 48 | 0 |
| u120_01 | 49 | 49 | 0 |
| u120_02 | 46 | 46 | 0 |
| u120_03 | 49 | 49.1 | 0.20 |
| u120_04 | 50 | 50 | 0 |
| u120_05 | 48 | 48 | 0 |
| u120_06 | 48 | 48 | 0 |
| u120_07 | 49 | 49 | 0 |
| u120_08 | 50 | 51 | 2 |
| u120_09 | 46 | 46.54 | 1.17 |
| u120_10 | 52 | 52 | 0 |
| u120_11 | 49 | 49 | 0 |
| u120_12 | 48 | 48.86 | 1.79 |
| u120_13 | 49 | 49 | 0 |
| u120_14 | 50 | 50 | 0 |
| u120_15 | 48 | 48 | 0 |
| u120_16 | 52 | 52 | 0 |
| u120_17 | 52 | 52.14 | 0.26 |
| u120_18 | 49 | 49 | 0 |
| u120_19 | 49 | 50 | 2.04 |

and u250_16, the ABO-1DCSP is between 0.06% and 0.99% above $N_{min}$ to fulfill item orders. While in the rest of the instances, the ABO-1DCSP is between 1%, and 1.06% above $N_{min}$.

In Table 14 it can be seen that the ABO-1DCSP reaches the $N_{min}$ in 7 of the 20 instances. Meanwhile, in the instances u500_02, u500_11, u500_13, u500_14 and u500_17 the ABO-1DCSP is between 0.03% and 0.22% above $N_{min}$. For the rest of the instances, the ABO-1DCSP uses between 0.43% and 0.51% more stock than $N_{min}$.

**Table 13  Comparison of $N_{min}$ stock used of instances u250 of *Falkenauer (1996)*.**

| Instance | $N_{min}$ | ABO-1DCSP | % ABO-1DCSP > $N_{min}$ |
|----------|-----------|-----------|--------------------------|
| u250_00 | 99 | 99.52 | 0.52 |
| u250_01 | 100 | 100 | 0 |
| u250_02 | 102 | 102.26 | 0.25 |
| u250_03 | 100 | 100 | 0 |
| u250_04 | 101 | 101.1 | 0.09 |
| u250_05 | 101 | 102 | 0.99 |
| u250_06 | 102 | 102 | 0 |
| u250_07 | 103 | 104.1 | 1.06 |
| u250_08 | 105 | 106 | 0.95 |
| u250_09 | 101 | 101 | 0 |
| u250_10 | 105 | 105 | 0 |
| u250_11 | 101 | 102 | 0.99 |
| u250_12 | 105 | 106 | 0.95 |
| u250_13 | 102 | 103.02 | 1 |
| u250_14 | 100 | 100 | 0 |
| u250_15 | 105 | 106 | 0.95 |
| u250_16 | 97 | 97.06 | 0.06 |
| u250_17 | 100 | 100 | 0 |
| u250_18 | 100 | 101 | 1 |
| u250_19 | 102 | 102 | 0 |

Table 15 shows that the ABO-1DCSP reaches the $N_{min}$ in 6 of the 20 instances. For the instances u1000_01, u1000_08, u1000_10, u1000_18, and u1000_19, the ABO-1DCSP generates solutions that require between 0.005% and 0.05% above $N_{min}$. Meanwhile, in the rest of the instances, the ABO-1DCSP needs between 0.11% and 0.42% more stock than $N_{min}$ to satisfy the demand for items.

In the same way we consider some instances of sets 1 and 2 of *Scholl, Klein & Jürgens (1997)* to corroborate the efficiency of ABO-1DCSP under the parameter stock used against the $N_{min}$ reported in *Gherboudj (2019)*. The instances consider a single type of stock that can be 100, 120 or 1,000. As for the items, the quantity of these is between 50 and 500 with lengths that can be from 2 to 494.

Table 16 shows the comparison of the ABO-1DCSP under the parameter of stock used against the $N_{min}$ of some instances of set 1 of *Scholl, Klein & Jürgens (1997)*. As seen in 11 out of 20 instances the ABO-1DCSP can reach $N_{min}$ to fulfill the demand of items. Meanwhile, in 7 instances the ABO-1DCSP is between 0.05% and 0.95% above $N_{min}$. Only in 2 instances the ABO-1DCSP is 1% above $N_{min}$.

Table 17 presents the comparison of some instances of set two of *Scholl, Klein & Jürgens (1997)*. It is observed that in fifteen out of twenty instances the ABO-1DCSP reaches $N_{min}$. While in four instances the ABO-1DCSP uses between 0.99% and 1.35% more stock than $N_{min}$. Only in one instance the ABO-1DCSP reach a percentage above 3% $N_{min}$.

**Table 14  Comparison of $N_{min}$ stock used of instances u500 of *Falkenauer (1996)*.**

| Instance | $N_{min}$ | ABO-1DCSP | % ABO-1DCSP > $N_{min}$ |
|---|---|---|---|
| u500_00 | 198 | 198.92 | 0.46 |
| u500_01 | 201 | 202 | 0.49 |
| u500_02 | 202 | 202.46 | 0.22 |
| u500_03 | 204 | 205 | 0.49 |
| u500_04 | 206 | 206 | 0 |
| u500_05 | 206 | 206 | 0 |
| u500_06 | 207 | 208 | 0.48 |
| u500_07 | 204 | 205 | 0.49 |
| u500_08 | 196 | 196.86 | 0.43 |
| u500_09 | 202 | 202 | 0 |
| u500_10 | 200 | 200 | 0 |
| u500_11 | 200 | 200.3 | 0.15 |
| u500_12 | 199 | 200 | 0.50 |
| u500_13 | 196 | 196.06 | 0.03 |
| u500_14 | 204 | 204.26 | 0.12 |
| u500_15 | 201 | 201 | 0 |
| u500_16 | 202 | 202 | 0 |
| u500_17 | 198 | 198.18 | 0.09 |
| u500_18 | 202 | 202 | 0 |
| u500_19 | 196 | 197 | 0.51 |

From the results of the instances tested above, the ABO-1DCSP shows to be consistent in regard to the $N_{min}$. For the instances of *Liang et al. (2002)*, the results do not exceed 4% above $N_{min}$. From the results of *Falkenauer (1996)* the ABO-1DCSP is around the 2% in one the u120 instances and below 2% for the rest of them. Similarly, for most of the instances u250 is below 1%, for instances u500 and u1000 is less than 0.5% and 0.25%, correspondingly. Finally, for the instances from *Scholl, Klein & Jürgens (1997)* is less than 2% for the set 1 and less than 4% for the set 2.

## CONCLUSIONS

In this work we presented the ABO-1DCSP, an adaptation of ABO to solve the 1D-CSP. The main idea is to utilize the advance of the herd of buffaloes to build a new best buffalo with a procedure of crossing with the best cutting patterns of the current herd before restarting it. The ABO is a swarm metaheuristic algorithm used in combinatorial problems like TSP and 1BPP, where the ABO demonstrated the capability and efficiency to obtain optimal solutions, remarking that 1BPP belongs to cutting and stock problems. We use a set of instances employed by other works based on exact, heuristic, or metaheuristic methods to compare the ABO-1DCSP under three parameters: waste, stock used, and stock used with waste. Also, we found that giving more time, specifically ten iterations, to the best buffalo of the herd to search for a better solution leads to better solutions without

**Table 15  Comparison of $N_{min}$ stock used of instances u1000 of *Falkenauer (1996)*.**

| Instance | $N_{min}$ | ABO-1DCSP | % ABO-1DCSP > $N_{min}$ |
|---|---|---|---|
| u1000_00 | 399 | 399.6 | 0.15 |
| u1000_01 | 406 | 406.24 | 0.05 |
| u1000_02 | 411 | 411.48 | 0.11 |
| u1000_03 | 411 | 412.74 | 0.42 |
| u1000_04 | 397 | 398 | 0.25 |
| u1000_05 | 399 | 399.78 | 0.19 |
| u1000_06 | 395 | 395 | 0 |
| u1000_07 | 404 | 404 | 0 |
| u1000_08 | 399 | 399.08 | 0.02 |
| u1000_09 | 397 | 398 | 0.25 |
| u1000_10 | 400 | 400.02 | 0.005 |
| u1000_11 | 401 | 401.96 | 0.23 |
| u1000_12 | 393 | 393 | 0 |
| u1000_13 | 396 | 396 | 0 |
| u1000_14 | 394 | 395 | 0.25 |
| u1000_15 | 402 | 403 | 0.24 |
| u1000_16 | 404 | 404 | 0 |
| u1000_17 | 404 | 405 | 0 |
| u1000_18 | 399 | 399.08 | 0.02 |
| u1000_19 | 400 | 400.02 | 0.005 |

constantly restarting the herd. We conducted experiments that help us to confirm the efficiency of the ABO-1DCSP algorithm in reducing waste. Furthermore, the ABO-1DCSP was able to obtain acceptable results for the stock used to satisfy the demand for items in the majority of the instances. Although, the objective function does not consider minimizing the number of stocks. The results obtained by the present investigation show that our approach generates on average, less or equal waste in 60% of the instances against heuristic, metaheuristic, and hybrid methods. Meanwhile, under the parameter of the stock used, our method generates solutions with an equal or lower average in 50% of the instances of *Liang et al. (2002)* against the approaches that were compared. In the same way, the ABO-1DCSP was compared against the $N_{min}$ of the instances of *Liang et al. (2002)*, where in 50% of the instances, the ABO-1DCSP reached the $N_{min}$. Meanwhile, the ABO-1DCSP is above 3.92% $N_{min}$ for the other instances of *Liang et al. (2002)*. Likewise, the ABO-1DCSP was also tested on four sets of *Falkenauer (1996)*, where the ABO-1DCSP reached the $N_{min}$ in 35 out of 80 instances, that is 43.75% of the instances. Meanwhile, the ABO-1DCSP kept a difference to the $N_{min}$ of less than 2.05%, with the rest of the instances. In the same way, the ABO-1DCSP was tested on some instances from *Scholl, Klein & Jürgens (1997)*, reaching the $N_{min}$ in 65% of them. Meanwhile, the ABO-1DCSP kept a difference of less than 2% with respect to the $N_{min}$ in most of the remaining instances.

With regard to the 1a to 10a instances, the comparison with the RPD values shows that the ABO-1DCSP performs better than the other methods to minimize waste.

**Table 16  Comparison of $N_{min}$ stock used of instances set 1 of *Scholl, Klein & Jürgens (1997)*.**

| Instance | $N_{min}$ | ABO-1DCSP | % ABO-1DCSP > $N_{min}$ |
|---|---|---|---|
| N1C1W1_A | 25 | 25 | 0 |
| N1C1W1_B | 31 | 31 | 0 |
| N1C1W1_D | 28 | 28 | 0 |
| N1C1W1_E | 26 | 26 | 0 |
| N1C1W1_F | 27 | 27 | 0 |
| N1C1W1_G | 25 | 25 | 0 |
| N1C1W1_I | 25 | 25 | 0 |
| N2C1W1_Q | 46 | 46.72 | 1.56 |
| N2C1W2_N | 64 | 64 | 0 |
| N2C1W2_O | 64 | 65.14 | 1.78 |
| N2C1W2_P | 68 | 68.04 | 0.05 |
| N2C1W2_R | 67 | 67 | 0 |
| N3C1W1_A | 105 | 106 | 0.95 |
| N3C2W2_D | 107 | 107.58 | 0.54 |
| N3C2W4_B | 112 | 112.8 | 0.71 |
| N4C1W2_T | 323 | 323.44 | 0.13 |
| N4C1W4_A | 368 | 368 | 0 |
| N4C1W4_B | 349 | 349.76 | 0.21 |
| N4C1W4_C | 365 | 365 | 0 |
| N4C1W4_D | 359 | 360.8 | 0.50 |

Meanwhile, the non-parametric Friedman test confirm that the differences with RPD values between all methods are significant. The second important finding was that solutions obtained by ABO-1DCSP required a quantity of stock close to or equal to the $N_{min}$, positioning in the second place among all methods corresponding to RPD values. Also, applying the test of Friedman ensured that the differences were significant. The third important aspect is related to the number of stocks used with waste produced from the arrangement of items obtained by the ABO-1DCSP was close in most instances. The ABO-1DCSP was placed in the second position according to RPD values.

Taking all together, it appears that the ABO-1DCSP developed in this article is an effective algorithm to solve the 1D-CSP. For future work, other issues could be addressed; for example, using a new objective function that considers both the number of stocks with waste and total waste. This can improve the search for solutions to use the fullest of each stock and minimize the total waste at the same time in the cases of 1D-CSP with single or multiple stocks. Thus, a local search method could be integrated with the ABO-1DCSP to obtain better solutions.

**Table 17  Comparison of $N_{min}$ stock used of instances set 2 of *Scholl, Klein & Jürgens (1997)*.**

| Instance | $N_{min}$ | ABO-1DCSP | % ABO-1DCSP > $N_{min}$ |
|---|---|---|---|
| N1W1B1R2 | 19 | 19 | 0 |
| N1W1B1R9 | 17 | 17 | 0 |
| N1W1B2R0 | 17 | 17 | 0 |
| N1W1B2R1 | 17 | 17 | 0 |
| N1W1B2R3 | 16 | 16 | 0 |
| N2W1B1R0 | 34 | 34 | 0 |
| N2W1B1R1 | 34 | 34 | 0 |
| N2W1B1R3 | 34 | 34 | 0 |
| N2W1B1R4 | 34 | 34 | 0 |
| N2W3B3R7 | 13 | 13 | 0 |
| N2W4B1R0 | 12 | 12 | 0 |
| N3W2B2R3 | 39 | 39.42 | 1.07 |
| N3W3B1R3 | 29 | 29 | 0 |
| N3W4B1R1 | 23 | 23 | 0 |
| N3W4B2R1 | 22 | 22.78 | 3.54 |
| N4W2B1R0 | 101 | 102 | 0.99 |
| N4W2B1R3 | 100 | 101 | 1 |
| N4W3B3R7 | 74 | 75 | 1.35 |
| N4W4B1R0 | 56 | 56 | 0 |
| N4W4B1R1 | 56 | 56 | 0 |

### Funding

This study was supported by the National Council of Humanities Sciences and Technologies (CONAHCYT) for financial support through scholarship number 770376 and projects with numbers CB- 2017-2018-A1-S-43008 and F003-320109. The funders had no role in study design, data collection and analysis, decision to publish, or preparation of the manuscript.

### Grant Disclosures

The following grant information was disclosed by the authors:
The National Council of Humanities Sciences and Technologies (CONAHCYT): CB-2017-2018-A1-S-43008, F003-320109.

### Competing Interests

The authors declare there are no competing interests.

### Author Contributions

- Leonardo Javier Montiel-Arrieta conceived and designed the experiments, performed the experiments, analyzed the data, performed the computation work, prepared figures and/or tables, authored or reviewed drafts of the article, and approved the final draft.

- Irving Barragan-Vite conceived and designed the experiments, performed the experiments, analyzed the data, performed the computation work, authored or reviewed drafts of the article, and approved the final draft.
- Juan Carlos Seck-Tuoh-Mora conceived and designed the experiments, analyzed the data, prepared figures and/or tables, authored or reviewed drafts of the article, and approved the final draft.
- Norberto Hernandez-Romero conceived and designed the experiments, analyzed the data, prepared figures and/or tables, authored or reviewed drafts of the article, and approved the final draft.
- Manuel González-Hernández conceived and designed the experiments, analyzed the data, prepared figures and/or tables, authored or reviewed drafts of the article, and approved the final draft.
- Joselito Medina-Marin analyzed the data, performed the computation work, prepared figures and/or tables, authored or reviewed drafts of the article, and approved the final draft.

## Data Availability

The data is available at Zenodo: LeoMontielArrieta. (2023). LeoMontielArrieta/ABO-1DCSP: ABO-1DCSP (1.0). Zenodo. https://doi.org/10.5281/zenodo.8361112.

## Supplemental Information

Supplemental information for this article can be found online at http://dx.doi.org/10.7717/peerj-cs.1728#supplemental-information.

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
