# Peer review of "Minimizing the total waste in the one-dimensional cutting stock problem with the African buffalo optimization algorithm"

_PeerJ Computer Science, doi:10.7717/peerj-cs.1728_

## Round 0.1 · original submission · Major Revisions

Please go carefully over all the reviewer comments and address each. Please make sure that the experimental part is robust with respect to benchmarks and previous work on the problem.

Reviewer 1 ·

Basic reporting

Reviewer Report on Manuscript
Minimizing the total waste in the one-dimensional cutting stock problem with the African Buffalo Optimization Algorithm
by Leonardo Javier Montiel-Arrieta, Irving Barragan-Vite, Juan Carlos Seck-Tuoh-Mora, Norberto Hernandez-Romero, Manuel Gonzalez-Hernandez, and Joselito Medina-Marin

General comments

The paper addresses the one-dimensional cutting stock problem (1D-CSP). The 1D-CSP is a widely studied optimization problem that involves determining how to cut a set I = {1,…, m} of item types from an unlimited source of large objects of length L. Each item type i \in I is characterized by a length l_i and demand d_i. The objective is usually to minimize the number of large objects used to fulfill the demand for items. The 1D-CSP arises in several manufacturing systems such as in the cutting of paper reels and steel bars. The paper develops an African Buffalo Optimization (ABO) algorithm, which is a bio-inspired meta-heuristic, to solve the 1D-CSP. In addition, it presents computational results seeking to evaluate the performance of the approach in comparison with benchmark approaches from problem instances of the literature.

Major comments

1) A major point of concern is the use of English. There are several errors easily detected by a non-native English speaker like me. I list some of them below. I recommend that the paper should be proofread by a professional editing service.
2) Another point of concern is the literature review. In my opinion, this section presents a collection of papers from the literature without a clear organizational structure or a critical discussion of the literature. There are two basic unanswered questions: (a) what are the literature's state-of-the-art heuristic/meta-heuristic approaches to the 1D-CSP? (b) what are the most used sets of benchmark instances and their features? In contrast, the present paper uses three long paragraphs citing various references to the detriment of "educating" the audience about related work. In my opinion, the answers for topics (a) and (b) should be revisited in the section of computational experiments as benchmark approaches and benchmark datasets, respectively.
3) Notice that the 1D-CSP and the one-dimensional bin packing problem (1D-BPP) are two related cutting problems that differ only concerning the data. The items are aggregated into item types in the 1D-CSP (there is a demand for each item type), while the items are not aggregated in the 1D-BPP (so, the demand for an item is unitary). The paper claims that there is an ABO algorithm for the 1D-BPP. So, why is not this algorithm used to solve the 1D-CSP instead of developing a new ABO algorithm for the 1D-CSP? Moreover, why is not this algorithm taken as a benchmark approach?

Experimental design

4) A last major point of concern is the section of computational experiments. I strongly disagree with the presented argument that there are no consensus benchmark instances in the literature for evaluating approaches to 1D-CSP. It is very difficult to evaluate the performance of an approach to a widely studied combinatorial problem such as the 1D-CSP from just 10 benchmark instances. For instance, I recommend using as a dataset the 18 classes generated by Gau and Wäscher (1996) problem instances, which were used in Cerqueira et al. (2021) as mentioned in the section of the literature review. Other widely used benchmark datasets are available at <https://www.euro-online.org/websites/esicup/data-sets/>.
5) Moreover, it is known from the literature that minimizing the number of objects used and minimizing material waste represent the same objective when the demand for items must be met strictly, that is, without shortages or excesses. However, the authors use these two metrics to imply that they are different things. Note that in Equation (2) on Page 5 the demand for items is strictly met; the explanation of the proposed algorithm also hints at this. So why were these additional metrics used?

Validity of the findings

It would be important to validate the results obtained against benchmark instances from the literature with a known optimal value. In this sense, it would be possible to know whether the solutions obtained are optimal, near-optimal or of poor quality.

Additional comments

Minor comments

6. (Abstract, line 13) Please replace “consists in” with “consists of”.
7. (Abstract, line 15) Please avoid the term “and the like”.
8. (Introduction, lines 43-44) How to measure the effectiveness and efficiency of an algorithm?
9. (Introduction, lines 61-65) Please see the use of quotes. Revise that in other parts of the text.
10. (Introduction, line 67) There are references in parentheses in the text that should be in the normal style.
11. (State of the art of 1D-CSP, line 68) Please define the term “LP” before using it.
12. (State of the art of 1D-CSP, line 81) Would it be “Constraint Programming”?
13. (State of the art of 1D-CSP, line 94) Please revise the sentence.
14. ( State of the art of 1D-CSP, line 125) Please revise the sentence.
15. (Description of 1D-CSP) Please revise Model (1)-(2) to include the feasibility of the cutting patterns.
16. (Description of 1D-CSP, line 235) Please refer to equations using parenthesis, that is, Equation (1) and Equation (2). Revise that in other parts of the text.
17. (The African Buffalo Optimization Algorithm, line 242) Please insert a space between the sentences.
18. (The African Buffalo Optimization Algorithm, lines 246-248) Please define “bgmax”.
19. (Experiments and results) Please insert the caption on the top of the tables.
20. (Experiments and results, Table 2) Please provide a metric with the relative length of the item types in relation to the length of the large object.

Annotated reviews are not available for download in order to protect the identity of reviewers who chose to remain anonymous.

Reviewer 2 ·

Basic reporting

The “state of the art” section has large paragraphs, some with more than 20 lines. It would be better for the reader that these paragraphs be divided into smaller ones.

Please correct reference “Evtimov, G. and Fidanova, S. (2018). Ant Colony Optimization Algorithm for 1D Cutting Stock Problem. Springer International Publishing. https://doi.org/10.1007/978-3-319-65530-7_3”.

It seems that the problem formulation given in Eqs. (1) and (2) is based on compact formulation, but I think that it is lacking some knapsack-type constraints. You can take a look at https://doi.org/10.1061/(ASCE)CO.1943-7862.0000 for an example of a compact formulation for the 1D-CSP.

Experimental design

There are some flaws in experimental design. The paper reports results from computational experiments with the ABO algorithm. It uses a set of 10 benchmark instances found in the literature. The size of the dataset is too small to draw general conclusions about the relative performances of competing algorithms. I recommend that the authors carry out experiments with 500 hundred instances or more of varying sizes. In addition, it is not clear if the authors reimplemented each one of the tested algorithms or just copied the results from papers in the literature. The problem with just taking results from the literature instead of implementing is that some algorithms will have different implementations. These will be an additional source of error and may favor some algorithms with better implementations.

Moreover, the total number of function evaluations a heuristic algorithm makes is critical in its performance. The total number of function evaluations is a proxy for the computational effort of a heuristic algorithm to achieve a given performance. Ideally, all algorithms should be compared for the same number of function evaluations. If this is not possible, at least they should be run for the the same total time in the same machine. It is very hard to draw general conclusions regarding alternative heuristic algorithms if these factors are not controlled.

Validity of the findings

I think findings are compromised due to methodological flaws pointed out in the Experimental Design section above. For example, the authors claim that “The comparison with the RPD values demonstrate that the ABO-1DCSP was the best method to minimize waste”, but as stated above, the computational experiments are insufficient to draw such a strong conclusion.

·

Basic reporting

Clear and unambiguous, professional English used throughout.

Experimental design

Research question well defined, relevant & meaningful. It is stated how research fills an identified knowledge gap.

Methods described with sufficient detail & information to replicate.

Validity of the findings

Review Report

I have gone through the content of the paper entitled “Minimising the local waste in one dimensional cutting stock problem with the African buffalo optimization algorithm” and found that the content of the paper is interesting and concise but there are the following reasons for not accepting as such:

1 Since we know that cutting stock problem is a practical problem for real industry and authors claim that develop method is competitive in waste minimization on the basis of only 10 instances taken from the Literature.
2 Line number 245 where bgmax is not define properly.
3 Line number 250 written that the value of Lambda is low. But it is not explained that how much low in numerically or some other sense.
4 In description of the ABO 1DCSP, Method or rule for arrangement of the items is not shown in Figure 1. Also, how these patterns are created and what is the procedure to form the buffaloes are not defined.
5 In the subsection ABO 1D CSP, what is the value of bgmax in sense of 1DCSP and what is the formula to evaluate the value of bgmax. Also, it is not clear that how patterns are formulated in the algorithm.
6 How can decide the number of buffaloes in the iteration and how this object length is related to 1DCSP terminology in algorithm line number 1.
7 Also, in the algorithm part line number 8, what do you mean by i%q==0 in sense of 1DCSP.
8 In the line number 322, how this updateBuffaloes function is related the process of cutting pattern.
9 In the section of experiments and results, data 3a, 4a instances and 8a, 9a instances have the same values then how they give different output also there is no need to check or run the algorithm in 4a and 9a instances data as mention in Table 2.
10 Also in above section, authors have calculated RPD but it is not clear that for which data set and how authors have calculated average RPD in Table 6.
11 ABO 1DCSP is giving similar results but not minimum as comparison to other methods shown in the table 5.
12 Existing methods are giving similar result as proposed method ABO 1DCSP is giving. Even in some cases results calculated by ABO 1DCSP are used more average stock than existing method as shown in Table 7. Then, what is the advantage to use ABO 1DCSP method.
13 The proposed method is giving same average stock used with waste in Two instances out of 10. Authors claim that method is completive but experimental data taken from literature giving better results as compared to proposed method.
14 In order to show the competitiveness of the proposed, authors should add some practical instances.
15 If Authors rectify above suggestions, then proposed method may lead to enhance the knowledge and subjective progression of 1DCSP Literature.

Final Comment:
Manuscript cannot accept in this form. It may be accepted after revision as per above suggestions.

Additional comments

Manuscript cannot accept in this form. It may be accepted after revision as per above suggestions.

---

## Round 0.2 · accepted · Accept

Two of the previous reviewers have reviewed your revision and are happy with the way you addressed the issues. Based on their comments no further technical changes are required for it to be published.

Reviewer 1 ·

Basic reporting

In this revised version, the authors met all the points that I suggested revision in the previous version. In my opinion, the paper can be accepted for publication in its present form.

Experimental design

The computational experiments were improved with the inclusion of new benchmark problem instances.

Validity of the findings

The results are solid and accurate.

Reviewer 2 ·

Basic reporting

The authors improved general presentation of the paper, according to the recommendations in the previous review report.

Experimental design

The authors corrected many of the flaws detected in the initial submission. In particular, they increased the number of problems instances tested from 10 to 120.

Validity of the findings

The findings of the authors are in line with what is already known of most meta-heuristic algorithms. The ABO algorithm is very similar to other bio-inspired algorithms, such as ant colony optimization, whale optimization, and so on. Due to this similarity, it is expected that its performance would be similar. Anyway, these results reinforce the hypothesis that most bio-inspired algorithm are just variations of random search with additional operators. The choice of which algorithm to use will be based largely on familiarity and personal preference.